

# Precipitation-fire-functional interactions control biomass stocks and carbon exchanges across the world's largest savanna

Mathew Williams[1,2,*], David T Milodowski[1,2], T Luke Smallman[1,2], Kyle G Dexter[1], Gabi C Hegerl[1], Iain M McNicol[1], Michael O'Sullivan[3], Carla M Roesch[1], Casey M Ryan[1,2], Stephen Sitch[3] and Aude Valade[4]

[1] School of GeoSciences, University of Edinburgh, EH9 3FF, UK

[2] National Centre for Earth Observation, University of Edinburgh, EH9 3FF, UK

[3] Faculty of Environment, Science and Economy, University of Exeter, EX4 4QF, UK

[4] Eco&Sols, Univ Montpellier, CIRAD, INRAE, Institut Agro, IRD, Montpellier, France

* Corresponding author: mat.williams@ed.ac.uk



## 1 Abstract

Southern African woodlands (SAW) are the world's largest savanna, covering ~3 M km$^2$, but their carbon balance, and its interactions with climate and disturbance are poorly understood. Here we address three issues that hinder regional efforts to address international climate agreements: producing a state-of-the-art C budget of SAW region; diagnosing C cycle functional variation and interactions with climate and fire across SAW; and evaluating SAW C cycle representation in land surface models (LSMs). Using 1506 independent 0.5$^\circ$ pixel model calibrations, each constrained with local earth observation time series of woody carbon stocks ($C_{wood}$) and leaf area, we produce a regional SAW C analysis (2006-2017). The regional net biome production is neutral, 0.0 Mg C ha$^{-1}$ yr$^{-1}$ (95% Confidence Interval –1.7 - 1.6), with fire emissions contributing ~1.0 Mg C ha$^{-1}$ yr$^{-1}$ (95% CI 0.4-2.5). Fire-related mortality driving fluxes from total coarse wood carbon ($C_{wood}$) to dead organic matter likely exceeds both fire-related emissions from $C_{wood}$ to atmosphere and non-fire $C_{wood}$ mortality. The emergent spatial variation in biogenic fluxes and C pools is strongly correlated with mean annual precipitation and burned area. But there are multiple, potentially confounding, causal pathways through which variation in environmental drivers impacts spatial distribution of C stocks and fluxes, mediated by spatial variations in functional parameters like allocation, wood lifespan and fire resilience. Greater $C_{wood}$ in wetter areas is caused by positive precipitation effects on net primary production and on parameters for wood lifespan, but is damped by a negative effect with rising precipitation increasing fire-related mortality. Compared to this analysis, LSMs showed marked differences in spatial distributions and magnitudes of C stocks and fire emissions. The current generation of LSMs represent savanna as a single plant functional type, missing important spatial functional variations identified here. Patterns of biomass and C cycling across the region are the outcome of climate controls on production, and vegetation-fire interactions which determine residence times, linked to spatial variations in key ecosystem functional characteristics.

Key words: SAW, Southern Africa, LAI, land surface models, fire, vegetation carbon



## 2 Introduction

Tropical savannas, dominated by trees and grasses, cover 40% of the vegetated tropics (Pennington et al., 2018) including 2.3-3.1 M km$^2$ in southern Africa (Ribeiro et al., 2020;Ryan et al., 2016). Savanna C stocks and net C fluxes are substantial in the global carbon cycle (Sitch et al., 2015), but with major geographical variations. Spatially there is a strong coupling between precipitation and tree cover across African savanna, particularly where annual precipitation is < 800 mm (Sankaran et al., 2005). The presence of substantial, dry fuel loads means that disturbance from fire is common during the dry season (Andela et al., 2017). Fire influences decadal C sinks through combustion related emissions (van der Werf et al., 2017) and disturbance impacts on both vegetation growth rates (Yin et al., 2020) and tree mortality (Levick et al., 2015). Overall, the interactions of climate and disturbance, particularly from fire, generate dynamic conditions for C stocks and fluxes across tropical savannas and woodlands (Archibald et al., 2013;Lehmann et al., 2014), which are poorly mapped and understood.

Southern African woodlands (SAW) are the dominant land cover in the dry tropics of southern Africa (Campbell, 1996), and form the world's largest savanna (Mistry, 2014;Ryan et al., 2016), covering much of Tanzania, Mozambique, Zambia, Zimbabwe, Malawi, Angola and southern DRC. The woodlands of this region are phylogenetically distinct from other tropical savannas (Dexter et al., 2015) and have biogeochemical and fire patterns (Alvarado et al., 2020) that are linked to unique functional traits (Osborne et al., 2018). These woodlands have long been subjected to, and thus are highly adapted to, disturbance by people, fire (generally set by people), and herbivores (Chidumayo, 2002;Chidumayo, 2004). Overall, the woodland C cycle is often non-steady-state, and anthropogenic change is strengthening this tendency (Ryan et al., 2016). Fire impacts on the C cycle and vegetation C stocks are linked to wet seasons moist enough for biological production to generate fuel load, and dry seasons intense enough to dry fuel for destructive fires. Wetter areas of the SAW region may have biomass stimulated by rising production but limited by rising mortality from fire.

A complete ecosystem C cycle analysis for the SAW region, that spans climatic gradients, resolves process interactions between climate, fire and the ecological functioning of C cycling, does not currently exist. There are knowledge gaps both on biosphere-atmosphere exchanges and on internal ecosystem processing of C. Deriving dynamics of C requires quantification and linkage of relevant processes controlling the biosphere-atmosphere exchange of C, its allocation or transfer to different C pools, and the turnover of these pools. Eddy flux data are scarce and short term in this region (Merbold et al., 2009). As a result, the net biome exchange (NBE) of $CO_2$ and its components (e.g. gross primary production (GPP), ecosystem



respiration ($R_{eco}$), fire emissions ($E_{Fire}$)) remain poorly quantified (Ciais et al., 2011;Ernst et al., 2024).
Internal C processes, particularly mortality or turnover of key pools (linked to mean residence time, MRT),
are critical for determination of C balance but poorly quantified (Friend et al., 2014;Smallman et al., 2021).
The MRT is the ratio of C pool size to the total losses from that pool per unit time. In savanna, MRT is
sensitive to both external factors like burning and to internal ecosystem properties. External factors like
burning are likely to shorten residence times, but vegetation may adapt to burning with increased tissue
resilience to fire.  Plant tissue (wood, foliage) lifespans may vary spatially, for instance with climate.
These C cycle knowledge gaps hinder national efforts to manage savanna carbon stores to meet international
actions like the Paris Agreement of the UNFCCC. Also, these gaps weaken model projections of trajectories
of C for this region under climate change. Simulation models typically represent tropical woodlands across
the globe using a single 'plant functional type' (PFT), with PFT-specific parameters which may lead to
biased outcomes (Bloom et al., 2016). The functional differences within the savanna biome (Lehmann et
al., 2014;Moncrieff et al., 2014) mean that region-specific carbon cycle estimates linked to locally valid
functional characteristics are required. Even within the SAW region, we expect to find biological variation
and gradients in functional characteristics (Osborne et al., 2018). Understanding this variation and links to
the environment can underpin more robust knowledge. This knowledge can improve representation and
therefore forecasts from land surface models, for instance those used to study trends in the land carbon
cycle, such as the Trendy experiment (Sitch et al., 2015).
Insights into SAW C cycling are accumulating through intensive studies and extensive observations.
Researchers have developed robust methods for woodland inventory and landscape sampling (SEOSAW
partnership, 2021). Chronosequence studies have documented the biomass recovery rates of these
ecosystems post-disturbance (Chidumayo, 2004;Chidumayo, 2013;Kalaba et al., 2013;Gonçalves et al.,
2017) to provide insights into annual to decadal dynamics. Earth observations (EO) of vegetation greening
(changes in leaf area index, LAI) have been found reliable against *in situ* data on canopy phenology (Ryan
et al., 2014;Ryan et al., 2017) and hence can map potential for photosynthesis in time and space. Radar
remote sensing has been identified as an effective tool for mapping biomass and its changes over these
landscapes (Ryan et al., 2012;Mitchard et al., 2009). These actions have developed the first regional
analyses for biomass in space and time (McNicol et al., 2018;McNicol et al., 2023). Long term observations
from satellites track the burned area across these landscapes (Chuvieco et al., 2019). These multiple new
analyses of the SAW region provide an opportunity to generate a more robust assessment of the C cycle
from local to regional scales. Mechanistic models calibrated with these data can provide a complete,
constrained, and probabilistic quantification of the carbon cycle and its processes.





In the present study, we combine new spatial data products with a model-data fusion system (CARDAMOM
(Bloom and Williams, 2015)), to create the most comprehensive diagnostic analysis to date of the $CO_2$-C
cycle of the SAW region in southern Africa. We use this analysis to address questions about key controlling
processes on the dynamics of major C pools, and their variation with climate and fire disturbance across
the region for 2006-2017. We further characterise net $CO_2$ exchanges resulting from different driving
factors and variations in plant processes, including allocation and mortality. Net ecosystem exchange (NEE
= $R_{eco}$ – GPP; sink has a negative sign) is purely biogenic, i.e. biological processes driven by atmospheric
conditions. Net biome production (NBP) includes human-driven emissions from prescribed factors such as
fire and land use removals (NBP = – NEE – fire emissions – biomass removals by external factors; sink has
a positive sign). Specifically, this study generates a full C cycle analysis and asks the following research
questions (RQ):

13       1.   How do fluxes and resulting net exchanges of $CO_2$ vary across the SAW region and covary with

14            climate, fire, and functional characteristics?

15       2.   How do carbon stocks and their longevity covary with climate, fire, and functional characteristics?

16       3.   How does data-constrained analysis of ecosystem C cycling compare to Trendy land surface model

17            estimates for the region?

For RQ1 we hypothesise that biogenic fluxes (GPP, $R_{eco}$) will be determined by a positive relationship with
precipitation, the dominant control on biological metabolism in SAW (Campbell, 1996). We hypothesise
that NBP across SAW will be determined by a negative relationship with burned area, through fire
emissions ($E_{Fire}$). For RQ2 we hypothesise that C stocks in total coarse wood C ($C_{wood}$) will be positively
correlated with, and their distribution determined by, precipitation. But we hypothesise there will be
mediating effects from variations in functional characteristics such as wood lifespan and fire resilience,
evidenced by broad scale gradients in these ecosystem functional characteristics. For RQ3 we hypothesise
that comparisons of land surface models from Trendy with CARDAMOM analyses will be more consistent
in biosphere-atmosphere fluxes than in stock estimates, because of the challenge of calibrating modelled
stocks to observations (Fawcett et al., 2022).
The novelty of this research is threefold. The regional C budget produced here is state-of-the-art due to its
consistency with locally calibrated estimates of woody biomass dynamics from earth observation. Causal
inference approaches disentangle emergent spatial patterns in C dynamics and ecosystem functional
characteristics, providing new biogeographical understanding of ecological functioning and diversity. The
spatially detailed model calibration builds an emergent map of process and C cycle variation that allows
resolution of within biome patterns, enhancing assessment of LSMs.





## 3 Methods

Multiple EO products of C stocks and LAI, and a soil C map, are combined into a pixel-by-pixel regional
analysis, through assimilation with an intermediate complexity biophysical ecosystem model (Bloom and
Williams, 2015) that is calibrated over the area of interest (Figure 1) with local climate, fire and forest
clearance forcing data. The result is a rigorous, probabilistic C cycle assessment, including GPP, NBP,
allocation to tissues, pool sizes, ecosystem processes, fire emissions, fire mortality and non-fire mortality.
Calibrated parameters and C cycle assessments are produced independently for each of the 1506 model
pixels at 0.5° spatial and monthly temporal resolution for a 12-year period (2006-2017 inclusive). The study
domain comprises all of Tanzania, Mozambique, Zambia, Zimbabwe, Malawi, Angola and southern
Democratic Republic of Congo (DRC) and covers 4.5 M km$^2$, including miombo woodland and a mix of
other woodland and savanna types and land uses (SEOSAW partnership, 2021;Godlee et al., 2021).
Statistical analysis then relates the spatially independent, data-consistent analytical outputs of each pixel to
climate, fire/human disturbance and to outputs of LSMs to address the research questions.

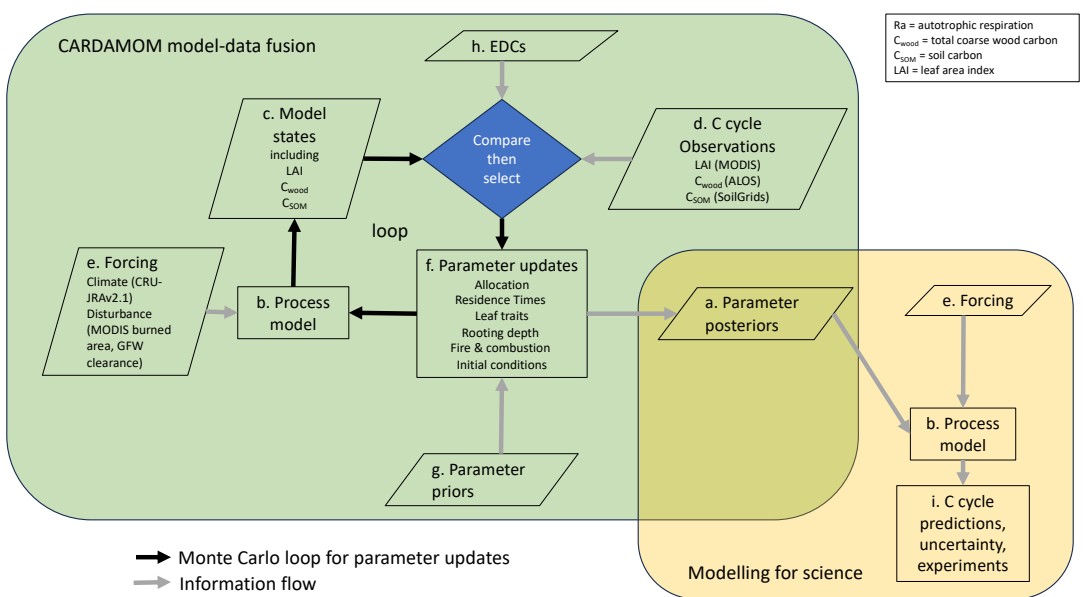

19 Figure 1. Schematic of the CARDAMOM methodology (green box) and modelling process (yellow box).

20 The Carbon Data Model Framework (CARDAMOM) generates parameter estimates with uncertainty (a)





for a process model (b). Independent estimates are made for each location (pixel) in the analysis. Parameter
estimates are constrained to ensure that specific model state variable predictions (c) match independent
observations for those variables at that location (d). Model predictions are made using local forcing data on
climate and disturbance (e). The model has 32 parameters (f) that govern biological processes, fire impacts
and include 7 initial conditions, with priors provided for each (g). A Monte Carlo process explores
parameter space defined by the priors, comparing model estimates (c) with observations (d), and using
ecological and dynamical constraints (EDCs, h) to inform selection (accept/reject) of parameter
combinations. Once parameter posterior ensembles are generated for each pixel (a), then a separate
modelling process uses these parameters to generate ensemble C cycle estimates for each pixel (i) using the
model (b) and specified forcing (e).
**3.1 Environmental data**
*3.1.1 Biomass, LAI time series and soil C data for calibration*
25 m resolution L-band radar data from ALOS-PALSAR were used to estimate aboveground woody carbon
(AGC), based on a calibration with field estimates (McNicol et al., 2018). We used a scalar linking above
and belowground wood C stocks ($C_{wood}$ = 1.42 x AGC (Ryan et al., 2011)) to prepare four annual 0.5° maps
of $C_{wood}$ for the 4-year period 2007-2010 based on higher resolution data from McNicol et al. (2018).
Uncertainty in the biomass observations (2.5 tC ha$^{-1}$) was estimated based on a local characterisation of
bias in retrieved biomass (McNicol et al., 2018).
MODIS EO (Myneni et al., 2021) product number MCD15A2H.061 provided 8-day composite information
on LAI (2006-2017) aggregated to months. Prior information on soil carbon stocks to a depth of 1.0 m were
drawn from the SoilGrids2 database (250 m resolution), a machine-learning based interpolation of field
inventories (Hengl et al., 2017). All data were aggregated to the 0.5° model spatial grid resolution. LAI and
soil carbon estimates were provided with a corresponding uncertainty estimate from their respective
products. The assimilation makes uses of LAI data available for all months of the analysis (n=144), biomass
data for four of the 12 years (n=4), and soil C data as a single value applied to its initial status (n=1).
*3.1.2 Disturbance and burned area observations for driving analyses*
MODIS product number MCD64A1.061 provided monthly, 500 x 500 m burned area data (Giglio et al.,
2018). Tree cover loss is imposed as a fractional removal of biomass, derived from the 30-m resolution
Global Forest Watch data on area disturbed (Hansen et al., 2013). Both data sets were aggregated to the
model 0.5° spatial grid and monthly resolution. Land use change or vegetation transition was not included
in the dynamics of the modelled ecosystem.



*3.1.3 Woody biomass chrono-sequences for model validation*

Chronosequence data provided estimates of the accumulation rate of woody biomass for two areas in the SAW region. At N'hambita, Mozambique, we generated estimates of biomass from 28 plots each of 0.125 ha, with age since abandonment ranging from 2-30 years (Williams et al., 2008). At Kilwa District, Tanzania, we used estimates from 55 plots each of 0.2 ha, with age-since-abandonment of 2-47 years (McNicol et al., 2015).

*3.1.4 Meteorological and soil physics data for model forcing and soil parameters*

CARDAMOM meteorological drivers were extracted from the CRU-JRAv2.1 dataset, a 6-hourly 0.5º dataset of precipitation using the Japanese Reanalysis product (see (Harris, 2019)) and aggregated to monthly resolutions (Figure S 1). Soil sand/clay fractions required for estimating soil hydraulic properties for input to the ecosystem model in CARDAMOM are extracted from the SoilGrids2 dataset.

## 3.2 Modelling the carbon cycle

*3.2.1 Terrestrial Ecosystem Model*

An intermediate complexity ecosystem model, DALEC-4 (Williams et al., 2005), simulated carbon stored in both live biomass (labile, foliage, fine roots and total coarse wood which includes stems, branches, and coarse roots) and dead organic matter (a litter pool, and a Soil Organic Matter (SOM) pool that includes coarse wood debris). The model simulates C flows (allocation and turnover/mortality) between pools and with the atmosphere (photosynthesis and respiration) and requires 25 parameters and 7 initial conditions (Table 1). Processes are sensitive to climate drivers, and pools are sensitive to disturbance drivers (fire and other biomass removal). Photosynthetic uptake (GPP) is estimated by the Aggregated Canopy Model, ACM2 (Smallman and Williams, 2019), as a function of temperature, solar radiation, atmospheric $CO_2$, precipitation and LAI (LAI is simulated by DALEC). Water supply to the canopy is generated by a coupled water cycle model which estimates ecosystem water stock and accessibility as a function of precipitation, soil texture and wood and root C stocks. Autotrophic respiration ($R_a$) is estimated as a fixed fraction of GPP. Net primary production (NPP = GPP – $R_a$) is allocated using fixed fractions to live pools. Heterotrophic respiration of litter and soil carbon ($R_h$) is estimated as a function of carbon stock, a turnover rate and a temperature coefficient. Ecosystem respiration ($R_{eco}$) is the sum of $R_a$ and $R_h$. Canopy phenology is simulated by a model with pixel-specific fixed times each year for budburst and leaf senescence. Bud burst leads to allocation of C from the labile to foliar pool. Leaf senescence initiates turnover of C from the foliar pool. There is no explicit separation of tree and grass components in the model.





| Parameter | Prior low | Prior high | Units | Posterior to prior ratio | Parameter type |
|---|---|---|---|---|---|
| Decomposition rate | 0.00001 | 0.01 | $d^{-1}$ | 0.88 | res |
| Fraction of GPP respired | 0.2 | 0.8 | fraction | 0.61 | all |
| Fraction of NPP to foliage | 0.1 | 0.5 | fraction | 0.63 | all |
| Fraction of NPP after labile allocation to roots | 0.1 | 0.8 | fraction | 0.83 | all |
| Leaf Lifespan | 1.001 | 6 | y | 0.09 | fol |
| TOR wood | 0.000009 | 0.001 | $d^{-1}$ | 0.53 | res |
| TOR roots | 0.001368 | 0.02 | $d^{-1}$ | 0.90 | res |
| TOR litter | 0.0001141 | 0.02 | $d^{-1}$ at 0°C | 0.94 | res |
| TOR SOM | 0.000001368 | 0.00009126 | $d^{-1}$ at 0°C | 0.82 | res |
| temperature factor, Q10 | 0.019 | 0.08 | - | 0.93 | res |
| Canopy efficiency | 10 | 100 | $gCm^{-2}d^{-1}$ | 0.23 | fol |
| Leaf onset day | 365.25 | 1461 | Day of year | 0.12 | fol |
| Fraction of NPP after leaf allocation to $C_{lab}$ | 0.01 | 0.5 | fraction | 0.55 | all |
| $C_{lab}$ release period | 10 | 100 | d | 0.68 | fol |
| Leaf fall onset day | 365.25 | 1461 | Day of year | 0.03 | fol |
| Leaf fall period | 20 | 150 | d | 0.48 | fol |
| LCA (leaf C per area) | 20 | 180 | $gCm^{-2}$ | 0.75 | fol |
| IC $C_{lab}$ | 1 | 2000 | $gCm^{-2}$ | 0.03 | init |
| IC $C_{fol}$ | 1 | 2000 | $gCm^{-2}$ | 0.13 | init |
| IC $C_{root}$ | 1 | 2000 | $gCm^{-2}$ | 0.20 | init |
| IC $C_{wood}$ | 1 | 30000 | $gCm^{-2}$ | 0.02 | init |
| IC $C_{litter}$ | 1 | 2000 | $gCm^{-2}$ | 0.13 | init |
| IC $C_{SOM}$ | 200 | 250000 | $gCm^{-2}$ | 0.03 | init |
| IC soil water as fraction of field capacity | 0.5 | 1 | fraction | 0.84 | init |
| Fraction of $C_{wood}$ which is coarse root | 0.15 | 0.5 | fraction | 0.94 | root |
| Coarse root biomass to reach 50 % of max rooting depth | 100 | 2500 | $g\ m^{-2}$ | 0.82 | root |
| Max rooting depth | 0.35 | 20 | m | 0.83 | root |
| Biomass resilience to fire | 0.01 | 0.99 | fraction | 0.62 | fire |
| Combustion completeness for foliage | 0.01 | 0.99 | fraction | 0.73 | fire |
| Combustion completeness for root and wood | 0.01 | 0.99 | fraction | 0.24 | fire |
| Combustion completeness for soil | 0.01 | 0.1 | fraction | 0.58 | fire |
| Combustion completeness for litter | 0.01 | 0.99 | fraction | 0.90 | fire |



Table 1 Parameters for the DALEC model, showing their prior and posterior values for a selected
location, units, and the ratio of the posterior 95% confidence interval to the prior range. Parameters are
categorised according to their role in C dynamics as follows: Allocation (all), residence times (res), foliar
traits (fol), rooting depth (root), fire and combustion (fire) and initial conditions (init). TOR is turnover rate.
IC is initial condition. $C_{lab}$ is labile C pool that supports leaf flushing.
Fire emissions are determined from the fraction of each pixel burned multiplied by a combustion fraction
parameter from Exbrayat et al. (2018). Specific combustion parameters are applied for each C pool. Of the
non-combusted vegetation pools in the burned fraction, fire mortality moves a fraction of C to the SOM
pool, using a resilience parameter common to all tissues. The SOM pool is assumed to include coarse woody
debris (CWD), and simulated fire emissions from the SOM pool therefore include the contribution from
CWD. A fraction of the litter pool is converted to SOM because of fire. For biomass removals linked to
land use, C losses are determined by the fraction of each pixel deforested as identified by GFW forcing
data, with all foliage C transferred to litter pools, and 80% of aboveground wood biomass removed from
the ecosystem (i.e. human extraction). Other pools are not deemed affected by this disturbance.
*3.2.2 Calibration using model-data fusion*
CARDAMOM is a model-data fusion framework (MDF) which combines local observations, their
uncertainties and ecological knowledge of the terrestrial C cycle to calibrate DALEC parameters
probabilistically. CARDAMOM uses a Bayesian approach within an Adaptive-Proposal Markov Chain
Monte Carlo (AP-MCMC) algorithm to retrieve ensembles of local parameters for each 0.5° pixel,
consistent with local observations, uncertainties, climate and disturbance forcing, and ecological theory
embedded in DALEC's structure (Bloom et al., 2016).
All DALEC parameters have a specified prior range to guide calibration (Table 1). Specific prior estimates
(i.e. mean + uncertainty) are provided based on literature studies for (i) the fraction of GPP allocated to $R_a$
($R_a$: GPP = 0.46+/-0.12 (Waring et al., 1998;Collalti and Prentice, 2019)) and (ii) the canopy photosynthetic
efficiency ($C_{eff}$ = 21.1 +/- 8.5 (Kattge et al., 2011)). CARDAMOM imposes ecological realism, or common
sense, on parameter retrievals using ecological and dynamic constraints, EDCs. EDCs set the likelihood of
a given parameter proposal to 0 if none of the conditions defined by the EDCs are met. The EDCs are
intended to prevent three kinds of ecologically inconsistent parameter proposals: 1) unrealistic
combinations, e.g. to ensure that turnover of fine roots is faster than for wood (in the absence of
disturbance), 2) maintaining emergent ecosystem ratios within observed ranges, e.g. fine root to foliar ratio,
3) preventing inappropriate carbon stock dynamics such as exponential carbon stock changes on short time



scales outside disturbance/fire. The resultant DALEC parameter uncertainty encompasses the combined
uncertainties of the observational constraints, parameter priors, the prior ranges and the plausible ecological
parameter space as defined by the EDCs.
*3.2.3 Validation against independent regional products*
Once calibrated probabilistically at each pixel, DALEC is then run using the same forcing data to generate
local ensembles of C cycle estimates (Figure 1). The first stage of validation tests the calibration process
by evaluating the simulated LAI, $C_{wood}$ and soil C against the assimilated data for these variables to test for
an unbiased estimate and for spatial coherence (random error across pixels) for each variable. The second
stage of tests is to evaluate the CARDAMOM analyses against other regional products. For NBE the
reanalyses are compared against an ensemble of Carbon Tracker Europe (CTE) estimates (Koren, 2020);
for GPP against the combined estimates from FluxCOM (Jung et al., 2020), Copernicus (Fuster et al., 2020)
and FluxSatv2 (Joiner and Yoshida, 2021); and for fire emissions against the combined estimates of
GFEDv4.1s (van der Werf et al., 2017) and GFAS (Kaiser et al., 2012). The third stage of validation uses
two SAW locations with chronosequence data. The local 0.5º DALEC calibration from the analysis was
used in an experiment, with 90% of woody biomass removed in the model, and regrowth followed over
decades using historical climate data and burned area data.
**3.3 Trendy Model Analysis**
18 process-based Land Surface Models (LSMs) were applied in the "Trends and Drivers of Regional Scale
Terrestrial Sources and Sinks of Carbon Dioxide" (Trendy-v11) project that supported the Global Carbon
Budget 2022 assessment (GCB2022; (Sitch et al., 2015;Friedlingstein et al., 2022)). LSMs are applied in a
set of factorial simulations using forcing datasets of observed global $CO_2$ content, observation-based
merged climate forcing from CRUJRA and historical Land-Use and Land cover changes (LULCC)
(Friedlingstein et al., 2022). For the TRENDY v11 experiments, LSMs are typically applied at 0.5-degree
resolution over the period 1700 to 2021. A subset of LSMs include prognostic fire models (Table S1). We
analysed the simulation results from the 'S3' simulation, where all three drivers vary, for the period 2006-

27  2017.

To compare data-constrained estimates of the terrestrial C cycle for the region against the Trendy ensemble,
we assess the agreement between domain-aggregated estimates for key C stocks and fluxes and their
seasonality. We also provide an indication of the spatial-temporal consistency of each LSM with our
CARDAMOM benchmark based on the fraction of pixels (in space and time) for which each LSM estimate
falls within the CARDAMOM 95% confidence interval. The outputs of the analysis are also evaluated
against the mean of the Trendy ensemble for the region, and against individual models using spatial





statistics and temporal analysis of seasonal dynamics of net exchanges (NBP) and their component
processes ($R_a$, $R_h$, $E_{Fire}$).

### 3.4 Spatial carbon cycle variability and determinants

The simulated C dynamics reflect the responses of the ecosystem model within a multivariate driver and
data space. At an individual 0.5º pixel, the model structure and retrieved parameter values determine the
temporal C cycle response to the environmental drivers. However, across the model domain, parameters
are retrieved independently for each pixel, generating an emergent map of functional variation over SAW.
This approach is an alternative modelling paradigm to the approach used by LSMs for which a single set
of model parameters is used to represent a particular plant functional type. The biogeographic gradients in
the C stocks and fluxes across the SAW determined by our analysis therefore represent the combination of
effects and interactions between the spatial variability in environmental drivers and the spatial variability
in ecological function, as characterised by the retrieved variations in model parameters.
To understand and explore the spatial sensitivity of the C cycle and ecological processes to environmental
factors we used a causal analysis approach similar to previous empirical studies that have synthesised
multiple observation streams to understand biogeographic gradients and their relationship to environmental
drivers (e.g. (Lehmann et al., 2014)). Common with these observation-based studies, our retrieved
biogeographic gradients are not determined by a prior spatial model. However, the model-data fusion
approach provides some key benefits, notably: (i) synthesising multiple observation streams (and
uncertainties) at the pixel level into an ecologically coherent and internally consistent representation of C
stocks and fluxes (Smallman et al., 2022), and (ii) explicitly partitioning the C dynamics along particular
process pathways, such as production, allocation and mortality, thus providing more detailed insights into
the functional variation across the SAW region.
We applied Wright's path approach (Runge et al., 2015;Wright, 1921, 1934) to estimate linear direct causal
effects that link the temporally averaged, ensemble-median C diagnostics to environmental drivers across
SAW. Wright's method only applies in the linear case. Here, the direct causal effect of a variable $X_i$ on a
variable $X_j$ is essentially quantified as the slope of the linear regression of $X_i$ on $X_j$, where any source of
confounding is removed prior to the regression. Environmental drivers that we considered in the causal
analysis include observed meteorological variables (e.g. precipitation, abbreviated as PPTN) and modelled
quantities (e.g. GPP), which were selected to resolve their causal effects on C fluxes and stocks and to avoid
confounding. To account for the influences of climate on fire activity and productivity limitations on fuel
availability, we also included burned area, which was causally linked to fire-related fluxes driving mortality,
combustion-related emissions, and post-combustion transfers between pools. To compare linear direct
causal effects across variables, variables were standardised prior to the analysis. The total causal effect of



$X_i$ on $X_j$ was then estimated as the sum of the products of all possible causal pathways from $X_i$ to $X_j$ (Wright,
1934;Runge et al., 2015). Note, that when we refer to causal effects in this work, these are standardised
linear direct causal effects. For more detail, see the supplementary information.

## 5  4 Results

### 6  4.1 Calibration and validation

The calibration process constrained model parameters to differing degrees (Table 1). Strongest constraints
were for initial conditions for C pools; foliar parameters related to leaf lifespan, leaf flush and fall;
combustion completeness for wood; and canopy efficiency (productive capacity). The weakest constraints
were for residence times for litter, roots and SOM, rooting depth parameters and most fire/combustion
parameters. The variation in constraint is consistent with proximity of parameters to assimilated data, thus
parameters connected to LAI and $C_{wood}$ are best constrained.
The calibrated model outputs explained much of the observed spatio-temporal variation in MODIS LAI
($r^2$=0.93) and ALOS biomass ($r^2$=0.99) and the spatial variation in soil C ($r^2$=0.97). Normalised root mean
square errors were for LAI = 0.17; biomass = 0.06; soil C = 0.04. The calibration bias was 6% or less in all
cases (regression slopes: LAI =0.94; biomass=1.01; soil C =1.01).
For NBE, CarbonTracker Europe suggests a close-to-neutral exchange, with uncertainty spanning zero
(Figure S 2), consistent with CARDAMOM estimates: 0.0 (95% CI -1.7-1.6) MgC ha$^{-1}$ y$^{-1}$. CARDAMOM's
median regional GPP estimate was 16.1 (CI 13.1-18.8) Mg C ha$^{-1}$ yr$^{-1}$, within the range of estimates from
the earth observation-orientated GPP products (Figure S 2). CARDAMOM's median fire emissions fell at
the lower end of the range of fire emissions products (Figure S 2) and its uncertainties were much larger
than the products' range.
At the locations in Mozambique and Tanzania, recovery of $C_{wood}$ in the model was consistent with data
(Figure 2). The uncertainty in the model accumulation rate (95% confidence intervals) was similar in
magnitude to the spread of biomass across the field inventories. Differences in burned area in the model
simulations, rather than climate, explain the higher steady-state $C_{wood}$ stock in the Tanzanian site.



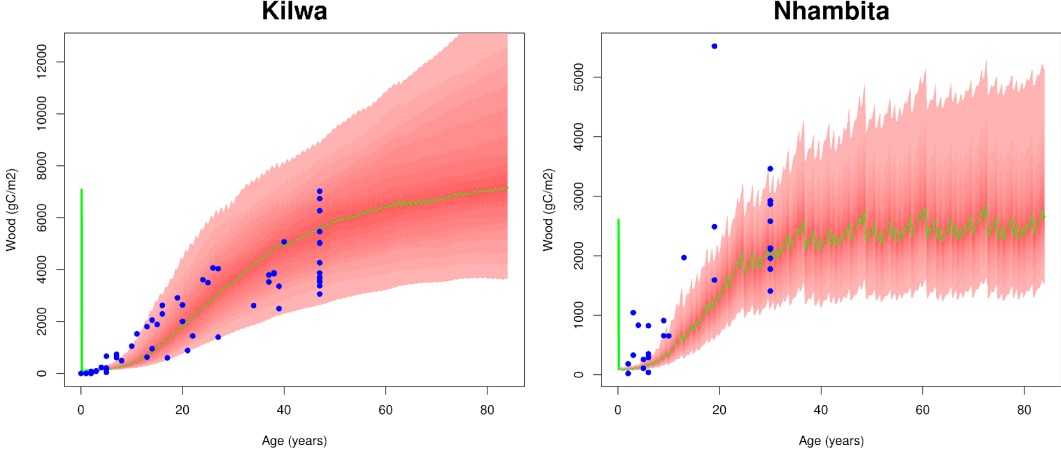

Figure 2. Independent test of wood biomass regrowth post-disturbance at two locations in southern African woodlands (left – Tanzania; right – Mozambique, note different scales). For both locations the DALEC model was calibrated at quasi-steady state using local EO data over the period 2006-2017 and local data on meteorology and burned area. 90% of wood steady state biomass was then removed (initial vertical green line at age=0) and modelled woody biomass accumulation (green line shows median, shaded interval shows 95% CI) is plotted against multiple independent chronosequence estimates based on data from fallow fields (blue dots).

**4.2 The carbon cycle of the SAW region**

CARDAMOM estimated that 49% of regional GPP is respired (Figure 3) and remaining NPP is allocated between foliage (median fraction = 0.18), a labile pool (0.13), fine roots (0.26) and $C_{wood}$ (0.37). Each ensemble member allocations sum to 1, but ensemble median fractions sum to < 1 (0.94) at the regional scale because posterior distributions of allocation in the analysis are not normal.

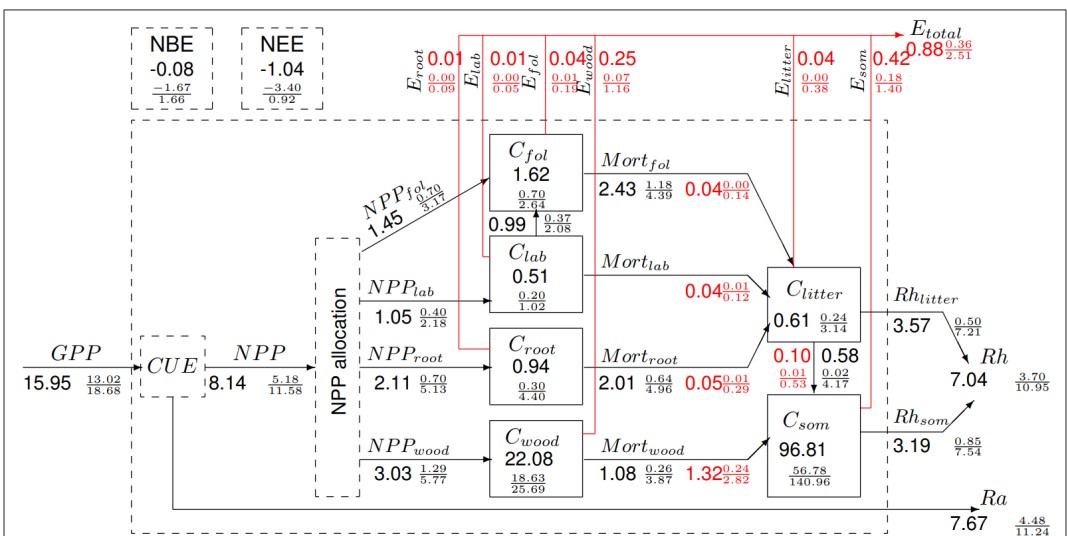

Figure 3. The C budget of the SAW region based on the CARDAMOM analysis at 0.5 x 0.5 degrees with a monthly time step between 2006-2017. Numbers show estimate of fluxes (alongside arrows) and of stocks (in boxes), using the mean value of all pixels in the SAW region. Units are MgC ha$^{-1}$ for stocks and MgC ha$^{-1}$ yr$^{-1}$ for fluxes. 95% confidence intervals are shown in a fractional form with 2.5 and 97.5 percentiles as numerator and denominator. Black fluxes are biogenic, including net primary production (NPP), mortality (Mort), autotrophic respiration (R$_a$) and heterotrophic respiration (R$_h$). NEE = R$_a$+R$_h$−GPP. NBE = NEE +E$_{total}$. Red disturbance fluxes are dominated by fire-driven emissions (E) and the fire-driven components of plant tissue mortality or loss of litter to SOM (indicated in red figures). Note that not all pools are in steady state and that the SOM pool includes coarse woody debris.

Mean residence times (MRT) of pools are sub-annual for foliage, labile, fine roots, and litter. MRT for wood is 8 years (95% CI 4-20 years) and for C$_{SOM}$ is 28 years (CI 11-90 years) (Figure S 3). Disturbance fluxes are 100-fold larger from fire rather than clearance (Figure S 1). On average 23% of the region's area is burned annually, mostly set by people. Burning losses from C$_{wood}$ are transferred to the atmosphere (~19% of total disturbance flux) or to dead organic matter (~81%). Losses from the C$_{wood}$ pool are largest through fire disturbance (~59% of total mortality flux) and remaining non-fire losses encapsulate pests, diseases, herbivory, plant aging, and degradation not detected by estimates of tree cover loss (Figure 3), but uncertainties are large. For other pools, both live and dead, non-disturbance flux magnitudes exceed disturbance fluxes. The regional C balance is approximately neutral (mean NBP: 0.0 (–1.7-1.6) Mg C ha$^{-1}$ y$^{-1}$). However, in the absence of fire disturbance (i.e. NEE), the region is a potential sink of 1.0 Mg C ha$^{-1}$ yr$^{-1}$.




NBP is a function of changes to total plant biomass (sum of all live C pools, $C_{veg}$) and to dead organic
matter (litter plus soil organic matter C, $C_{DOM}$), which are dominated by the two largest pools, $C_{wood}$ and
$C_{SOM}$. The analysis of changes to $C_{veg}$ ($\Delta C_{veg}$) is constrained by the assimilation of multiple biomass maps
2007-2010 (Figure 4), with largest losses in the east (Tanzania and N Mozambique) and through W Zambia
and S Angola. There are areas of positive $\Delta C_{veg}$ in S DRC, N Angola, E Zambia, W Zimbabwe and S
Mozambique. The distribution of $\Delta C_{veg}$ is unimodal and evenly distributed between regions of increasing
and decreasing $C_{veg}$ resulting in a regionally neutral stock change for $\Delta C_{veg}$ of 0.0 (-0.4/0.43) Mg C ha$^{-1}$ y$^{-}$
$^{1}$. The analysis of $\Delta C_{DOM}$ is not directly constrained by observations. $\Delta C_{DOM}$ is also unimodal, with a
relatively even split between areas accumulating and losing C from the soil. Uncertainties on $\Delta C_{DOM}$ are
approximately four times higher than for $\Delta C_{veg}$ (Figure 4).

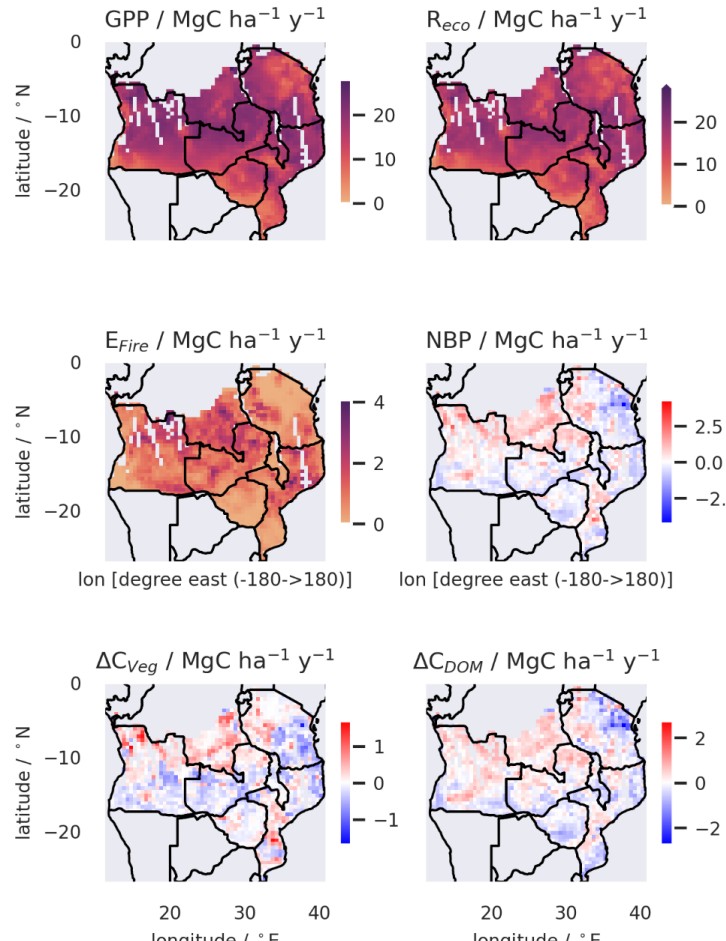



Figure 4. Spatial mapping of median gross fluxes, NBP, and temporally averaged rates of change in the live
pools ($C_{veg}= C_{wood} + C_{roots} + C_{foliage} + C_{labile}$) and dead organic matter ($C_{DOM} = C_{SOM}+C_{litter}$) C stocks across
the SAW region at 0.5º resolution, 2006-2017, as determined by diagnostic analysis. Gaps in maps relate
to areas without biomass observations due to gaps in ALOS-PALSAR data. GPP is gross primary
production; $R_{eco}$ is ecosystem respiration; $E_{Fire}$ is fire emissions; NBP = GPP – $R_{eco}$ – $E_{Fire}$ – biomass
removals by management (the latter are a relatively small flux compared to the others).
**4.3 Environmental controls on carbon fluxes (RQ1)**
Median GPP distribution across the SAW region (Figure 4) is skewed unimodal, with a peak at 20 MgC ha$^-$
$^1$ yr$^{-1}$ and a tail of lower GPP (Figure S 4). $R_{eco}$ is similarly skewed, and strongly spatially correlated (r=0.95)
with GPP, with a peak in its frequency distribution at 17 MgC ha$^{-1}$ yr$^{-1}$. Fire emissions fluxes ($E_{Fire}$) are
non-normal, dominated by low emissions (<1 MgC ha$^{-1}$ yr$^{-1}$) but with a tail of higher emissions up to 4
MgC ha$^{-1}$ yr$^{-1}$. The distribution of pixel-level median NBP peaks just below the source-sink boundary and
spans -2 to +3 MgC ha$^{-1}$ yr$^{-1}$. There is clear spatial structure to the fluxes, with higher GPP, $R_{eco}$, fire
emissions and NBP concentrated in certain areas (Figure 4) and correlated with forcings (Figure S 5).
The causal networks constructed to assess the controls on the spatial distribution of C fluxes identifies the
importance of precipitation and fire and their interactions (Figure 5, Figures S 6…S 8). Precipitation is the
dominant factor determining the rates of C cycling across the SAW, driving both the productivity and
mortality fluxes, with compensating effects on the overall C balance. Precipitation dominates the
distribution of GPP, with a standardised effect of 0.94 (0.90/0.98) [95% Confidence Interval]. Radiation is
positively linked to GPP (0.20; 0.16/0.24), while VPD (-0.13; -0.17/-0.11) and temperature are negatively
linked (-0.14; -0.17/-0.11). Precipitation is the dominant environmental driver of NPP (total standardised
effect: 0.86; 0.81/0.91), mediated by an environmental effect on carbon use efficiency (CUE). Precipitation
is also associated with the largest total standardised causal effects on the mortality fluxes driven by fire
(0.34; 0.31/0.38) and on non-fire mortality (0.55; 0.50/0.58). The total causal effect of precipitation on
gross fire mortality fluxes includes contributing causal pathways linked to the standing $C_{veg}$ stocks as well
as through influences on the fire-driven turnover of C (Figures S 7…S 9). Fire is a key source of C losses
in SAW woodlands. Burned area increases along the precipitation gradient (0.43; 0.37/0.48), and with
increasing VPD (0.34; 0.27/0.42). Burned area drives the fire mortality flux from the $C_{veg}$ pool (0.31;
0.28/0.33), with a significant mediating effect from the increasing resistance of C stocks to fire in fire-prone
areas described by spatial patterns in parameters (see Figure S 6).

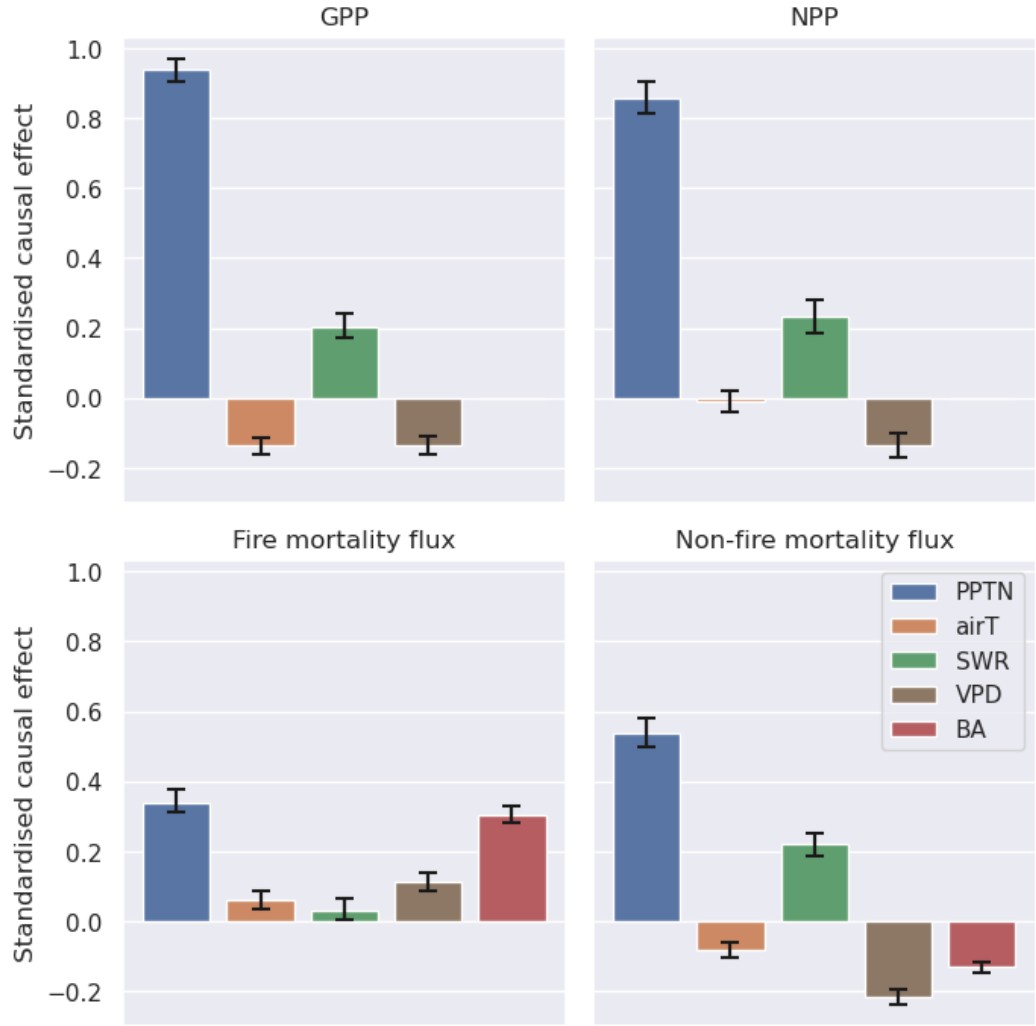

Figure 5 A summary of the causal effect analysis on spatial patterns in the pixel-median estimates of key
fluxes of C across the SAW region (with error bars for 95% bootstrapped CIs). Fluxes include GPP,
allocation to biomass (NPP), and mortality caused by fire and non-fire factors. For each flux the
standardised causal effects of different climate drivers (mean annual precipitation, PPTN; air temperature,
airT; short wave radiation, SWR; vapour pressure deficit, VPD) and fire (via burned area, BA) are
compared. Note that the causal analysis did not include a causal link between BA and GPP, NPP.



**4.4 Environmental controls on stocks and MRT (RQ2)**
C stocks in SAW are primarily in dead organic matter pools ($C_{DOM}$) with a mean of 98 MgC ha$^{-1}$ (95%
confidence internal, 57-142), 99% of which is $C_{SOM}$ to a depth of 1.0 m. Mean $C_{veg}$ are 26 MgC ha$^{-1}$ (22-
30), with 87% in $C_{wood}$. The mean ratio $C_{DOM}$:$C_{veg}$ is 4.0 (95% CI 2.1-12.5). Distributions of C stocks in live
and dead pools are unimodal (Figure S 9). The spatial patterns of C stocks are similar to the distributions
of biogenic fluxes (Figure 6).

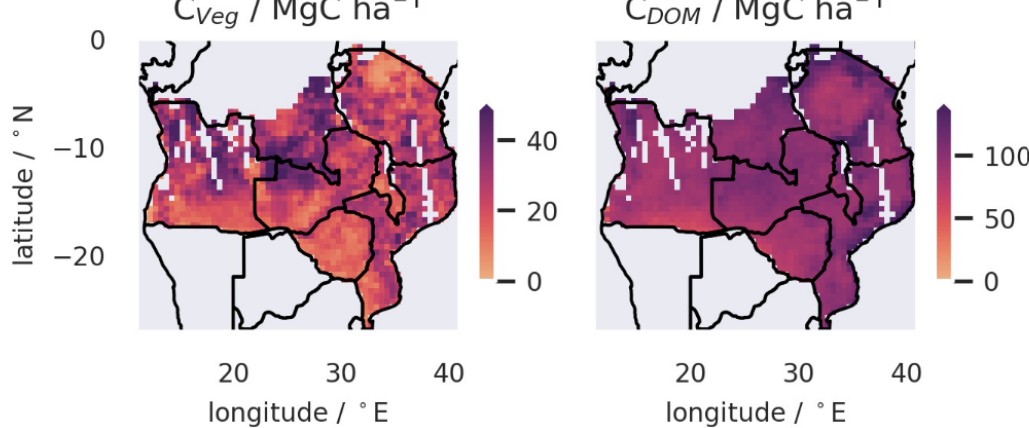

Figure 6. Spatial mapping of live C stocks, which are dominated by $C_{wood}$ (left) and dead organic C (right)
across the SAW region at 0.5° resolution, 2006-2017, as determined by diagnostic analysis. Gaps in maps
relate to areas without biomass mapping due to gaps in ALOS-PALSAR data.
The spatial distribution of C stocks depends on C assimilated via NPP and the rate of C turnover (T) (Figures
7, S 6). The spatial distribution of $C_{wood}$ is positively impacted by $NPP_{wood}$ (standardised effect 0.65;
0.61/0.69) and negatively impacted by turnover rates ($T_{wood,fire}$: –0.60; –0.67/–0.54; $T_{wood,other}$: –0.54; –0.58/–
-0.51). Causal analysis (Figure S 6) across the spatial dataset indicates that precipitation (PPTN) impacts
$C_{wood}$ along three mediating pathways: (A) positively via primary production (total effect of PPTN mediated
by $NPP_{wood}$ = 0.36; 0.32/0.40), (B) negatively via fire mortality rates (total effect of PPTN mediated by
$T_{wood,fire}$ = –0.07; –0.10/–0.04), and (C) positively via non-fire mortality rates (total effect of PPTN mediated
by $T_{wood,other}$= 0.11; 0.08/0.14). The analysis revealed clear emergent spatial variations in key functional



characteristics across the SAW region (Figure 8) controlling each of these pathways, including the fraction
of NPP allocated to wood (A); the fire resistance of ecosystems (B, determined as biomass resilience to fire
× (1 - Combustion completeness for wood); see Table 1); and the non-fire median turnover rate of $C_{wood}$.
The productivity pathway (path A) is the dominant control on the distribution of $C_{wood}$ across the SAW
(total standardised effect of PPTN on $C_{wood}$ = 0.40; 0.35/0.47). The impacts on $C_{wood}$ of turnover driven by
fire ($T_{wood,fire}$) and non-fire ($T_{wood,other}$) are comparable, but opposing and spatially variable (Figure 8). In
higher precipitation areas the link between relative fire mortality and burned area is weakened by a strong
compensating effect of higher fire resistance of vegetation (Figure S7). The total standardised impact of
fire (burned area) on $C_{wood}$ is negative (–0.33; –0.37/–0.30). The impact of other meteorological drivers
(VPD, short-wave radiation and air temperature) on $C_{wood}$ are relatively weaker. Overall fire emissions
represent a major loss from the $C_{wood}$ pool (Figure 3), with burned area driving fire-related turnover rates
(total causal effect: 0.55; 0.48/0.62) and hence MRT. We conclude that representation of SAW by a single
plant functional type (PFT) approach misses important spatial functional variations in residence times and
fire resistance.
The turnover of the fine root and foliage C pools are dominated by the phenological turnover associated
with seasonal growth and senescence directly tied to the seasonality of rainfall (Figure S7, S8). This
turnover is linked to the temporally averaged meteorological drivers, although with relatively weak
standardised effects. Generally, turnover rates (1/MRT) of both pools are negatively impacted by annual
PPTN and VPD, while annual temperature and short-wave radiation (SWR) have a positive effect, although
there is no clear dominant term. There is a correlation between PPTN and SWR (Pearson's r = –0.51).
Higher MRT for roots and foliage in wetter areas suggests extended phenology both above and
belowground, and identify a further important functional variation within SAW that a single PFT approach
misses.

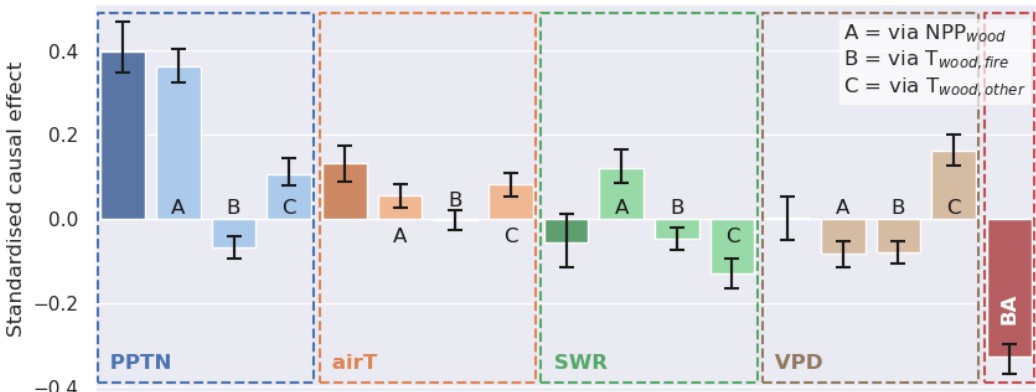



Figure 7 Summary of the causal effects from climate factors on spatial patterns in the pixel-median
estimates of total coarse wood C ($C_{wood}$) across the SAW region (with error bars for 95% bootstrapped CIs).
For mean annual precipitation (PPTN), air temperature (airT), short wave radiation (SWR), and vapour
pressure deficit (VPD), the total standardised causal effect is shown in the leftmost column of the four
panels. The three columns (A-C) show how the total effect for each factor is the outcome of three aggregated
causal pathways: climate effects operating through (A) changes to net primary production of wood, (B)
fire-driven turnover and (C) non-fire turnover. The total direct effect of fire (through burned area, BA) is
also shown for reference.

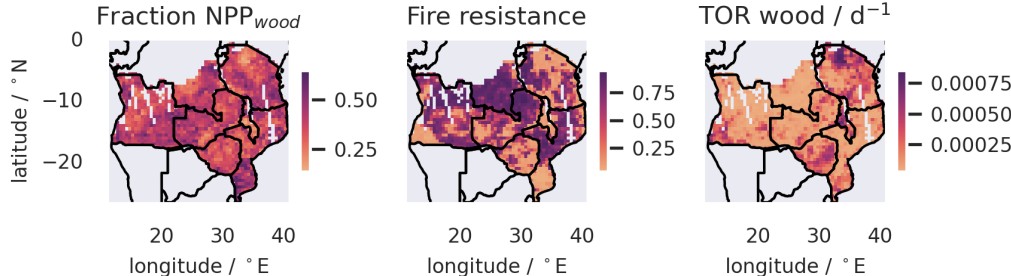

Figure 8. Spatial variations in three key ecosystem functional characteristics across Southern African
woodlands retrieved from the analysis. These three characteristics connect to the three pathways (Figure S
6) that are hypothesised to link spatial variation in environmental drivers (Figure S 1) to $C_{wood}$ (Figure 6).
Pathway (A) operates via variation in woody productivity, which is a function of the fraction of total NPP
allocated to wood, shown in the left panel; Pathway (B) operates through $C_{wood}$ turnover driven by fire,
which is linked to spatial variation in ecosystem fire resistance characteristics shown in the central panel;
and Pathway (C) is linked to variation in non-fire turnover rate (TOR), which has inferred spatial variations
as shown in the right panel.
**4.5 Comparison of observation-constrained analysis of C cycling to land surface model**
**estimates for the SAW region (RQ3)**
The seasonal cycles of GPP from CARDAMOM have similar amplitude and phase to the Trendy ensemble
mean, but individual Trendy models had larger variations in amplitude and phase, often outside the
CARDAMOM confidence interval (Figure S 10). For GPP, 13 of the 18 Trendy models had regional mean





annual estimates within the 95% CI of CARDAMOM estimates. The median annual GPP of the Trendy
ensemble (15.8 MgC ha$^{-1}$ yr$^{-1}$) was 2% less than the median CARDAMOM estimate (16.0 MgC ha$^{-1}$ yr$^{-1}$),
and comparable to the mean estimate for GPP of the independent observation-based products for the region
(15.7 MgC ha$^{-1}$ yr$^{-1}$) (Figure S 2). CARDAMOM NBP amplitude was larger than all but three of the Trendy
models, some of which had virtually no amplitude. These differences were linked to each major component
of emissions (Figure S 11).
The spatial overlap of GPP between the Trendy ensemble and CARDAMOM 95% CI was not complete,
ranging from 10% to 48% (Table S2; Figure S12-13), and typically lower during each wet season. For net
biome production, the mean estimates of all Trendy models were close to neutral over the region, consistent
with the CARDAMOM NBP. However, there were significant differences in amplitude and spatial
distribution (Table S1; Figure S13). The consistency of the spatial-temporal estimates of NBP for each
LSM with the CARDAMOM 95% CI ranged from 29% to 68% (Table S2; Figure S 14-15).
Estimates of $C_{veg}$ varied markedly between Trendy LSMs (15-66 MgC ha$^{-1}$) for the SAW region. Only three
out of 18 Trendy models had regional mean $C_{veg}$ estimates within the 95% CI of the CARDAMOM-DALEC
estimates (Table S1). The spatial distribution in $C_{veg}$ stocks varied markedly between LSMs (Figure S16-
17), with spatial-temporal consistency between individual LSMs and the CARDAMOM 95% CI varying
from 5% to 35% (Table S2), suggesting significant spatial biases. Considering the net change in the live
vegetation pools, $\Delta C_{veg}$, for which the CARDAMOM estimate is more closely constrained by the
assimilated data than NBP, the spatially coherent discord between the Trendy LSMs and the CARDAMOM
benchmark becomes more apparent (Figure 9, Figure S 17).

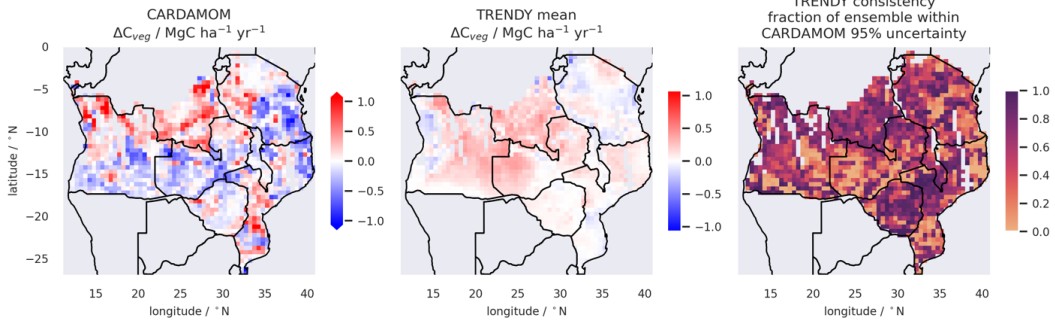

Figure 9. A comparison the data-constrained estimate of annual mean change in vegetation C stocks ($\Delta C_{veg}$)
from the CARDAMOM analysis with the mean estimate from the Trendy LSM ensemble. The right panel
shows the consistency of Trendy data by mapping the fraction of the 18 ensemble members with estimates



within the 95% confidence interval of the CARDAMOM analysis. Data cover the SAW region and the
period 2006-2017.
## 5 Discussion
### 5.1 Identification of carbon sinks and sources in the SAW region
The analysis reveals a balance between sources and sinks in this region from 2006 to 2017 (Figure 4),
dependent on the spatial gradients in productivity, driven by precipitation, and mortality, an important
component of which is driven by fire (Figure 6,7). Changes in $C_{veg}$ across the SAW have previously been
linked to varying patterns of land use and wood-fuel harvesting, and recovery of some woodlands with
reduced human pressures in other areas (McNicol et al., 2018). The explicit land-use flux modelled by
CARDAMOM is dependent on changes in tree cover detected by satellites, which indicated a small areal
extent of LUC forcing. Comparatively small disturbances typically associated with degradation processes,
e.g. wood-fuel harvesting, while potentially widespread (Bailis et al., 2015), are challenging to detect
(Milodowski et al., 2017) and maybe missed by the satellite products used in this analysis. Within the
CARDAMOM diagnostic analysis, C fluxes driven by non-fire degradation and not detected by GFW are
implicitly represented within the non-fire mortality flux, which contributes strongly to the spatial
distribution of $\Delta C_{veg}$. Development and assimilation of longer time series of wood biomass with low bias,
alongside robust time-series estimates of degradation, extent and intensity would help to refine
understanding of how anthropogenic activities impact the strength of the terrestrial C sink.
### 5.2 What are the environmental controls on exchanges of C throughout the region?
The analysis supported the hypothesis that precipitation has the dominant control on GPP across the region
(causal effect PPTN – GPP: 0.94; 95% CI: 0.90/0.98). This strong spatial relationship was the result of (i)
directly modelled links between soil moisture and stomatal conductance, and (ii) correlations between LAI
observational data (assimilated by CARDAMOM) and patterns of precipitation. Wetter areas were thus
associated with moister soils and higher LAI, both stimulating higher GPP, and indicative that water
availability is the principal limiting factor on GPP, consistent with (limited) eddy covariance data across
sub-Saharan Africa (Merbold et al., 2009).
We expected that productivity would positively impact burned area (BA), through fuel load (Fig S 6, S 7).
Our results were supportive to an extent (direct standardised causal effect of NPP on BA: 0.30; 0.21/0.38)
(Fig S 5), but burned area was also positively related to VPD (direct causal effect of VPD on BA: 0.38;
0.31/0.46), indicating that climate-dependent fuel moisture limitation may be as important as fuel load. Our
results are consistent with assessments that identified the SAW region straddling the transition between a



fire regime limited by fuel build-up and one limited by fuel moisture (Archibald et al., 2009a; Alvarado et
al., 2020; Archibald et al., 2009b).
We hypothesised that NBP across SAW would be negatively impacted by the burned area fraction. The
analysis supported this hypothesis: burned area was a strong driver of C losses; without the contribution of
fire emissions, the analysis indicated that the approximately C neutral SAW would have likely been a C
sink. However, burned area did not drive the spatial distribution of either $\Delta C_{veg}$ or NBP, due to concurrent
spatial gradients in NPP driven by precipitation (Figure 5), and mediating impacts across the SAW
environmental gradient arising from functional variations, including changes linked to wood lifespan and
effective fire resistance (Figure 8). As a result, despite constituting a major driver of C losses, burned area
fraction is actually positively correlated in space with NBP across the region (Pearson's r=0.28). The
emergent picture from the diagnostic analysis is that the carbon balance of the SAW region is determined
by the interplay between precipitation-driven gradients of productivity, and losses driven by a combination
of fire emissions and $R_h$, and that these fluxes are mediated by spatial variations in plant function linked to
climate gradients. The finding of function-climate gradients here matches plot level analysis along
precipitation gradients in West Africa (Zhang-Zheng et al., 2024).
The coarse spatial resolution of our analysis (0.5°) is unable to resolve the fine-scale heterogeneities in the
landscape. Grass litter is critical fuel for fires in the region (Archibald et al., 2009b), but our analysis does
not separate tree and grass foliage and litter pools. Our diagnostics indicated that the fire resistance of
vegetation increased with burned area, but secondarily also in wetter areas. These emergent responses could
be explained by direct plant-level adaptation to fire (e.g. thicker bark), or through community-level
feedbacks where fire is excluded due to increasing tree canopy cover excluding grass (Ryan and Williams,
2011; Ramo et al., 2021).

### 5.3 Controls on wood and soil C stocks

We hypothesised that C stocks in soils and biomass will be spatially correlated, and their distribution
determined by precipitation. Our analysis was supportive, with both stocks positively and most strongly
driven by precipitation (total causal effect: 0.40; 0.35/0.47), despite the mediating impact of precipitation
on burned area. Our analysis suggests that larger $C_{wood}$ stocks in wetter regions are sustained by a
combination of higher NPP and slower relative rates of turnover. Our hypothesis that $C_{wood}$ MRT is
inversely related to burned area is supported by the causal analysis (Figure S 6). Fire-related mortality from
$C_{wood}$ to $C_{SOM}$ likely exceeds fire-related emissions from $C_{wood}$ to atmosphere, and natural rates of $C_{wood}$
mortality fluxes into $C_{SOM}$ (Figure 3). Without fire disturbance, the MRT of $C_{wood}$ could more than double
from 8 to 20 years, and this would imply a similar proportional increase in steady state wood biomass,
increasing from a mean of 22 to 55 MgC ha$^{-1}$, a credible estimate based on fire exclusion experiments in
SAW (Ryan and Williams, 2011). Our conclusions for the dynamics of $C_{SOM}$ are necessarily weaker. We



lack robust constraint on $C_{SOM}$ dynamics, either though repeat mappings or through chronosequence studies.
Chronosequence data from part of the SAW suggest little change in soil C stocks after decades of post-
disturbance recovery.
We found support for our hypothesis that spatial variations in ecosystem functional characteristics influence
the distribution of biomass across SAW. The analysis revealed emergent regional gradients in ecosystem
functional characteristics related to woody allocation, wood lifespan and fire resilience (Figure 8), among
others. Analysis showed strong causal effects from climate and disturbance drivers on patterns of functional
variation (Figure S 6). Thus, wetter areas of the SAW tend to have live vegetation stocks with reduced
vulnerability to fire, longer wood lifespans in the absence of fire, and lower proportional allocation of NPP
to wood. There are also important functional variations in the dynamics of leaf and fine root pools linked
to climate, linked to strong phenological patterns across SAW (Ryan et al., 2017) and with impacts on
production patterns.
**5.4 Evaluation of Land Surface Models**
Our analysis supported the hypothesis that GPP and $R_{eco}$ fluxes from the Trendy models agree more closely
with CARDAMOM analyses than do Trendy models' estimates of C stocks (Table S1). Nevertheless, while
the domain aggregate estimates for GPP were comparable between Trendy mean and CARDAMOM
analyses, this obscures substantial variation among models (Table S1, Table S2), which showed strong
spatially structured variability inconsistent with CARDAMOM estimates (Figures S14, S17) (Teckentrup
et al., 2021). The apparent discrepancies highlight the challenges faced by the current generation of LSMs
to estimates the sensitivity of GPP to soil moisture variation in water-limited environments (Paschalis et
al., 2020;MacBean et al., 2021). There was greater disagreement between the Trendy ensemble and the
CARDAMOM estimate regarding $C_{veg}$ stock (Table S1, S2) and there were marked differences in their
estimates of the spatial distribution of $C_{veg}$ (Figure S12). On average, Trendy $C_{veg}$ across the SAW was
larger than CARDAMOM estimates (Table S1), in line with Trendy results over Australian savanna
compared with satellite estimates (Teckentrup et al., 2021) although this bias was not consistent across the
ensemble of LSMs.
Both Trendy models and CARDAMOM analyses suggest the region was close to neutral NBP. However,
Trendy models had lower seasonal variation in NBP than CARDAMOM. These differences were more
related to inconsistencies in C emissions from respiration and fire, rather than foliar phenology and GPP.
The low amplitude of NBP in Trendy models results from a strong temporal coupling in GPP and $R_{eco}$.
CARDAMOM analyses have large seasonal amplitudes arising from seasonal divergence, due to litter
production occurring at the end of the wet season, leading to dry season decomposition, coupled also with
dry season fires. The DALEC model lacks a soil moisture control on $R_h$, whereas most Trendy models do



include this relation. This structural difference may explain temporal differences in $R_h$ (Fig S 11),
particularly as the assimilated data have no direct constraint on $R_h$.

**5.5 Conclusions**

Our analysis reveals that carbon dynamics of the SAW are determined by the interplay between
precipitation and fire, mediated by substantial spatial variations in plant functional characteristics. Spatial
analyses from model-data fusion provided insights into SAW C dynamics variation in response to the
regional gradients in climate and disturbance. Precipitation is the dominant control on both primary
productivity (GPP) and C residence times. GPP variations are controlled directly by precipitation, through
soil moisture limitation on primary production, and indirectly through functional variations in phenology
(LAI). Precipitation gradients impact C residence times indirectly, through correlated variations in related
functional characteristics. For instance, precipitation is linked to patterns of effective fire resistance in
vegetation, and to variation in lifespan of $C_{wood}$ when fire is absent (Figure 8). Consequently, the spatial
distribution of C stocks across the SAW is significantly determined by the precipitation gradient through
multiple interacting pathways.
The full C cycle analysis of the region is the current state-of-the art due to its direct incorporation of repeat
biomass maps that are locally calibrated and validated. The analysis suggests that $C_{wood}$ mortality driven by
fire is attributed as the major loss term from $C_{wood}$, albeit with large uncertainties (Figure 3). The fire-driven
fall in $C_{wood}$ residence time across the precipitation gradient linked to rising burned area and fire mortality
(Figure 5), acts to damp positive feedbacks between increasing GPP and $C_{wood}$. If fire effects are removed,
our analysis suggest a ~3-fold increase in $C_{wood}$ (Bond et al., 2005). Much larger uncertainties remain in the
analysis of soil C due to sparsity of data compared to aboveground biomass.
This analysis has mapped variation in functional characteristics, challenging the use of a single PFT for this
region. CARDAMOM suggests substantial variations in functional characteristics across the SAW, for
instance for wood, foliar and fine root lifespans and allocation, and fire resistance. These variations likely
explain why LSM estimates are inconsistent with the data-constrained estimates from this study. Individual
LSMs deviated inconsistently from CARDAMOM estimates, with individual components of the C cycle
varying in space and between models. $C_{veg}$ stocks and fire emissions were the source of largest discrepancy,
alongside the temporal distribution of fluxes.
The C budgets here can also support more robust and observationally consistent national reporting in the
region for the Paris Agreement of the UNFCCC. The detailed resolution of the outputs, with locally valid
functional characteristics, can enhance national $CO_2$ emission factors for fire disturbance, for instance.
Working closely with national agencies, approaches such as demonstrated could deliver Tier 3 estimates of
national C budgets to support countries world-wide.



## 6 Acknowledgements

We thank Ben Poulter and Anthony Walker for their comments on the manuscript. We recognise UKRI grants to SEOSAW (NE/P008755/1), SECO (NE/T01279X/1), and NCEO. C Roesch and GH also thank the European Union's Horizon 2020 research and innovation programme under Marie Sklodowska-Curie grant agreement No. 860100 (iMIRACLI). We acknowledge and thank the broader CARDAMOM developer team. We thank the data providers from the Trendy LSM teams, MODIS teams, and SoilGrids community. C Ryan and IMM would like to thank JAXA for support via the EO-RA3 agreement no. ER3A2N035. The authors declare no conflicts of interest. This research was funded in whole, or in part, by NERC grants NE/P008755/1 and NE/T01279X/1. For the purpose of open access, the author has applied a creative commons attribution (CC BY) licence to any author accepted manuscript version arising.

## 7. Data Availability

The data that support the findings of this study are available in a resource at https://doi.org/10.7488/ds/7776. "Williams, Mathew; Milodowski, David Thomas; Smallman, Thomas Luke. (2024). Monthly Net Biome Exchange for the Southern African Woodlands 2006-2017 estimated using the CARDAMOM model-data fusion framework, 2006-2017. University of Edinburgh".

## 8. Author Contribution

MW, DTM and TLS conceived the analysis with support from CMRy, KGD and SS. DTM and TLS developed the model code and undertook the analysis with support from CMRo and GGH, and IMM, MOS and AV. DTM, TLS and MW produced visualisations. MW supervised the research and wrote the manuscript with input from all authors. MW, CMRy, KGD and SS provided funding for the work.

## 9. Competing Interests

The authors declare that they have no conflict of interest.

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
