# Peer review of "Precipitation-fire-functional interactions control biomass stocks and"

_EGUsphere, 2024_

## Author Response (AR2)

SAW paper, Dec 2024

Response to referees

Referee #1

Please provide more details about the method and materials, such as an accessible code of the model and data.

- We have added the DALEC code to the online data store for open access, alongside the analytical data outputs.

Also at Page 13, Lines 13-16: There are no materials to support these results.
- We have added a new Figure S 2, showing these results, to the supplementary information.

Page 13, Lines 17-18: Figure S2 does not include Carbon Tracker Europe.

- We recently updated the validation to OCO2 (replacing CTE) in Fig S2 – OCO2 is a more up-to-date product. We have updated the text to reflect this change in Fig S 2 (now Fig S 3).

Page 15, Lines 15-17: I cannot calculate the three numbers (19%, 81%, and 59%) based on the text and Figure 3. Could you clarify the calculations?
- The 59% is the calculation of total fire losses/all losses. i.e. $(1.32+0.25)/(1.32+0.25+1.08) = 0.59$
- The other figures are from an earlier analysis which we should have updated. We have changed the text to read ~16% and ~84%.

Page15, Lines 19-20: T The numbers do not match those in Figure 3, which may cause confusion. For instance, the mean value of NEE calculated as Ra + Rh - GPP = 7.67 + 7.04 - 15.95 = -1.24 does not correspond with -1.04 in Figure 3. The authors should verify all results carefully.

- This divergence results because the posterior ensemble distributions are non-normal. Thus, the median of the NEE estimates for each ensemble member produces a slightly different result to NEE estimated from the sum of the median Ra, Rh and GPP estimates for each ensemble member. So, the reported numbers are correct. We have clarified this issue in the caption for Fig 3. We had noted the issue of non-normality on P 14, L 15 of the submitted ms.

Additionally, the numerical representation in Figure 3 retains two decimal places, while other sections, like the abstract, use one decimal place. This inconsistency can be confusing; I recommend a unified format.

- We have selected and used a consistent number of decimal places throughout the manuscript for reporting C budgets and fluxes. We have updated the abstract to report two decimal places.

Page16. Lines 9-10: There appears to be a reference error; it may be Figure S4 instead of Figure 4. Additionally, how are the uncertainties measured from Figure S4? I cannot conclude that uncertainties on $\Delta C_{DOM}$ being four times higher than for $\Delta Cveg$.

- There is no reference error, Fig 4 is correctly referenced. We have clarified that the larger uncertainty is identifiable in Fig 4 *once the different scales are noted*. So, the new text can read "Uncertainties on $\Delta C_{DOM}$ are approximately four times higher than for $\Delta C_{VEG}$ (note different scales in Figure 4)."

Figure 4: Additional Y-axis titles in the second row need to be removed.
- We have made this adjustment.

Page 20, Lines 5-6: The text states that $T_{wood,fire}$ and $T_{wood,other}$ are spatially variable, but there is no spatial pattern for $T_{wood,fire}$ in Figure 8.
- We recognise our text was not clear on this point. Fig 8 does indeed show the spatial patterns as indicated, but uses different terms to title the panels. Thus $T_{wood,fire}$ is linked to the fire resistance parameter shown in the central panel of Fig 8. $T_{wood,other}$ is TOR wood in the right hand panel of fig 8. We have added to the text in the caption and section 4.4 to clarify the relationship of fire resistance shown in Fig 8.

Page 20, Line 2: A right bracket is missing.
We have made this adjustment.

Page 20, Lines 6-8: Is this referring to Figure S7? However, Figure S7 appears to relate to $C_{foliage}$ rather than $C_{wood}$.
This was an error; this should be a reference to Fig S6. We have adjusted, noting there is a new Fig S 2, so in the revised ms this reference is now to Fig S 6.

Referee #2

The description of fire emission calculations requires further elaboration and should clarify the following points without assuming in-depth familiarity with Exbrayat et al. (2018):
- How is "burnt area" used as a driver? Is there a specific step that converts burnt area into biomass or "vegetation pools in the burned area"?

We have adjusted the text to provide detail of how the burned area impacts on the C cycle, with specific equations:

$$E_x = B.K_x.C_x$$

For each model pixel, fire emissions from pool $x$ ($E_x$) are a function of pixel burned area fraction ($B$), a combustion completeness parameter for tissue type $x$ ($K_x$) and the C stock size of pool $x$. $K_x$ is calibrated by CARDAMOM.

*Equation 2*

$$M_{x,fire} = B.(1 - K_x).(1 - r).C_x$$

For the same pixel, fire-driven mortality of tissue $x$ ($M_{x,fire}$) is the uncombusted component of fire-impacted pool x, further modified by a vegetation resilience parameter $r$, also calibrated by CARDAMOM.

**Fire-driven mortality**

- On Page 10, Line 7, it is stated that emissions are calculated by multiplying burnt area by a combustion fraction parameter from Exbrayat et al. (2018). How does this differ from the "specific combustion parameters" applied to each C pool?

These are the same parameters. We have clarified the text.

- The same resilience factor is applied to all C pools (except SOM). How is this assumption justified?

We confirm that for each pixel a single vegetation fire resilience parameter is applied to all live pools. Across the 1506 analysed pixels, however, the independent pixelwise calibration results in a variation in the vegetation fire resilience parameter (Fig 8). We assume that vegetation in the SAW region displays characteristics distributed across a spectrum of fire resilience. Thus, some vegetation types have evolved to be resilient to fire, and others less so. Resilience is thus a holistic property of vegetation, rather than a tissue-specific property. Reflecting a common evolutionary history, fire resilience in vegetation is assumed to be similar across all vegetation pools within a pixel. Resilience is a property of vegetation.
The vegetation biomass resilience parameter will be constrained by the interactions between burned area and two vegetation observables - LAI dynamics and $C_{wood}$ dynamics. LAI data are more frequent and extend over a longer period that biomass data, so the strongest constraint on resilience will arise from the interactions between burned area observations and LAI observations. Thus, spatial patterns of fire resilience in the analysis will be inferred from the temporal interaction of the forest canopy (LAI) with burned area.
Combustion completeness is assumed to vary across pools within each pixel, reflecting how differences in structure, location and unit size of each vegetation pool affect combustion. These parameters also vary spatially across pixels.
We adjusted the text in section 3.2.1 to provide this explanation of these assumptions.

- Without a diagram, it is difficult to understand the flow of C from pool to emission and between pools, as described on Page 10, Lines 9–12 (although Figure 3 in the results section is somewhat helpful).

Figure 3 shows the model structure, and includes all pools and fluxes. We now reference this figure in Section 3.2.1 to provide more context.

Section 3.2.2 would benefit from an expanded explanation of how the fire and combustion parameters are constrained. Specific clarifications needed include:
- Is the only observational data available to constrain these parameters a rapid change in aboveground biomass/LAI, coinciding with a time step where significant burnt area is observed in the forcing?
Yes that is largely correct. However, even a small, sustained drop in biomass coinciding with burned area over a period provides information on fire impacts. We have adjusted the text in section 3.2.2 to clarify.

- Within the EDC framework, is there a mechanism to explicitly link biomass changes to fire occurrence, thereby impacting fire and combustion parameters? Alternatively, does the optimization infer fire, disturbance, and turnover parameters solely from biomass changes?
We confirm that EDCs do not relate explicitly to fire.
It is the repeat time series data on biomass that provides the key information on biomass change and therefore on the processes that drive change. Biomass observations are combined with (i) expected patterns of inputs to biomass (i.e. NPP, driven largely by LAI observations and climate, linked to the modelling of GPP) and (ii) outputs from biomass, linked to observed disturbance (from burned area and deforestation data), to infer the biomass dynamics.

- What happens if/when observations of rapid biomass changes and burnt area forcing do not align in space and time?
In this case, the calibration will adjust the non-fire mortality parameter in that pixel to match the biomass dynamics over the full analytical period. So, the overall trend in biomass change should be captured, but not attributed to fire. The analysis will not produce a biomass step change without an aligned forcing with burning or deforestation. We do note however, that operating at 0.5 degree resolution reduces the likelihood for temporal misalignment. We aggregate multiple burned area (500 m resolution) and biomass (25 m resolution) measurements to the analytical resolution (0.5 degrees).
We have updated the discussion in 5.1 to note this point.

- The simulated change in aboveground biomass appears to depend on both combustion completeness for wood and biomass resilience. What are the implications for estimating these parameters, which exhibit significant equifinality when constrained only by biomass change observations? How does this affect the fire-related Mort_wood flux and the subsequent C_som dynamics, leading to the large E_som flux?

This is an interesting question – it is true that our understanding of fire interactions is incomplete and challenged by the scales over which fire operates. As noted, biomass loss depends on both tissue combustion completeness (*K*) and vegetation biomass resilience (*r*). The biomass resilience parameter is constrained also by the interactions between burned area and LAI dynamics – not just woody dynamics. This constraint arises because resilience is a parameter that applies to all live pools (a property of vegetation), whereas combustion completeness is specific to each live pool. LAI data are more frequent and extend over a longer period that biomass data, so there is a stronger constraint on the resilience parameter from LAI. $K_{wood}$ is constrained largely by biomass observations, not by LAI. Thus, equifinality between *r* and *K* is reduced due to their differential constraint from independent observations. The massive ensemble calibration approach using multiple chains means that any equifinality is robustly characterised in the posteriors generated by the analysis. It is clear in Figure 3 that the uncertainty on the individual *Mort* fluxes and *E_som* fluxes are relatively large. We include discussion and emphasis of fire flux uncertainty in the discussion, section 5.2.

Section 4.1 provides limited information about how parameters, including fire and combustion parameters, are constrained or how model performance is validated.

We have added figure S2 to the SI showing the direct comparison of analytical outputs for LAI, biomass and soil C to the assimilated observations, to support the summary statistics presented in 4.1.
We refer to Table 1 which quantifies the constraint on each parameter relative to parameter priors. We could add further text in this section to summarise the degree of constraint shown in detail in the Table.
This section also refers to Figure 2 which is an independent validation of the calibrated model's capacity to simulate biomass accumulation over decades at two SAW sites.

To address this, the following questions should be clarified:
- On Page 13, Lines 20–21, it is stated that "fire emissions fell at the lower end of the range of fire emissions products," yet this appears to contradict the lower panel of Figure S2.
The text should read that the analysed median fire emissions were broadly "within the range of fire emissions products". The text has been updated.

- Page 13, Lines 21–22, mentions that "uncertainties were much larger than the products' range." Why is this the case, and what are the implications for interpreting emergent functional and causal relationships?
The calculation of regional fire emissions uncertainties from the CARDAMOM analysis is conservative. We assume that uncertainties are uncorrelated and so there is no effect of cancelling errors from combining multiple pixel analyses. In fact, there is likely to be some error correlation and therefore some reduction in errors at the regional scale. The analytical errors are large as a result of cascading uncertainties in C stocks in live and dead pools, and their interactions

with fire (combustion completeness). These uncertainties are directly tracked by the CARDAMOM approach and shown in the fluxes displayed in Figure 3.

The high uncertainty in fire-driven emissions is not an important factor in the interpretation of causal relationships because the causal analysis was confined to dynamics of live pools, and focused mostly on $C_{wood}$ dynamics, not emissions. This focus was deliberate to align with data availability (e.g. LAI and biomass observations) and thus rich information for calibration and inference of causation, using links to disturbance and climate data. There is no causal analysis of fire emissions or soil C dynamics. We adjusted the text in Section 3.4 to note these points.

Given the challenges in constraining fire and combustion parameters and the resulting large uncertainties in fire emissions, which hinder interpretation of ecosystem-level model outputs, it would be informative to compare CARDAMOM emissions with emissions from the prognostic fire models in the TRENDY LSM dataset. On Page 25, Line 30, it is suggested that "inconsistencies in C emissions from respiration and fire" account for mismatches in NBP, and this claim should be substantiated with more information on TRENDY model emissions in Section 4.5.

This comparison has been made in Figure S 11 (now S12), which shows the different patterns of fire emissions within the TRENDY ensemble, and TRENDY variations from the CARDAMOM analysis.

We have added a figure S12 reference at this point in the text to support our conclusion about inconsistencies.